

# Effects of size and microclimate on whole-tree water use and hydraulic regulation in *Schima superba* trees

Xiao-wei Zhao[1,2], Lei Ouyang[2], Ping Zhao[2] and Chun-fang Zhang[3]

[1] College of Life Sciences, Yulin University, Yulin, Shan Xi, PR China
[2] Key Laboratory of Vegetation Restoration and Management of Degraded Ecosystems, Guangdong Provincial Key Laboratory of Applied Botany, South China Botanical Garden, Chinese Academy of Sciences, Guangzhou, Guangdong, PR China
[3] Department of Life Sciences, Yuncheng College, Yuncheng, Shan Xi, PR China

## ABSTRACT

**Background**. Plant-water relations have been of significant concern in forestry and ecology studies in recent years, yet studies investigating the annual differences in the characteristics of inter-class water consumption in trees are scarce.

**Methods**. We classified 15 trees from a *Schima superba* plantation in subtropical South China into four ranks using diameter at breast height (DBH). The inter-class and whole-tree water use were compared based on three parameters: sap flux density, whole-tree transpiration and canopy transpiration over two years. Inter-class hydraulic parameters, such as leaf water potential, stomatal conductance, hydraulic conductance, and canopy conductance were also compared.

**Results**. (1) Mean water consumption of the plantation was 287.6 mm over a year, 165.9 mm in the wet season, and 121.7 mm in the dry season. Annual mean daily water use was 0.79 mm d$^{-1}$, with a maximum of 1.39 mm d$^{-1}$. (2) Isohydrodynamic behavior were found in *S. superba*. (3) Transpiration was regulated via both hydraulic conductance and stoma; however, there was an annual difference in which predominantly regulated transpiration.

**Discussion**. This study quantified annual and seasonal water use of a *S. superba* plantation and revealed the coordinated effect of stoma and hydraulic conductance on transpiration. These results provide information for large-scale afforestation and future water management.

Corresponding author
Ping Zhao, Zhaoping@scib.ac.cn

## INTRODUCTION

Plantation species that are artificially planted in rows in intensively managed stands have different ecological functions to that of natural forests; in particular their water use is significantly higher (*Farley, Jobbagy & Jackson, 2005*; *Nosetto, Jobbágy & Paruelo, 2005*; *Licata et al., 2008*). As sap flow technologies have become widespread, studies of whole-tree water relations have increased rapidly. Many of these studies focus on water consumption of planted trees, because of the potential risk to the water balance over a large area

(*Morris et al., 2004*; *Little et al., 2009*; *Rascher et al., 2011*). However, these studies focus more on inter-species than within-species comparisons.

China is the second largest plantation country in the world and has established extensive areas of plantations since the 1950s. The planting programs during the 1970s and 1980s mainly focused on fast-growing, high-yielding timber species, as well as ecosystem protection and rehabilitation (*FAO, 2010*; *FSC, 2012*). The rapid and large-scale simultaneous expansion of the size and growth of plantations caused concern in both researchers and the public over the balance of ground water, which was mainly concentrated in southern China and the Yellow River basin. Studies of plantation size focused on productive species containing fast-growing, economically valuable species (*Lane et al., 2004*; *Ma et al., 2008*; *Tan et al., 2011*; *Zhu et al., 2015*), while studies of plantation growth focused more on species of ecological recovery (*Ge et al., 2006*; *Cao et al., 2007*; *Cao, Chen & Yu, 2009*; *Chen et al., 2010*; *Wang et al., 2010*; *Zheng et al., 2012*; *Jian et al., 2015*). These studies drew different conclusions: that pure plantations: (1) posed a potential risk to ground water and were even considered as water pumps; (2) had a higher potential to reduce soil water storage in the watershed; and (3) posed no risk to ground water, but had potential hydrologic responses to climate and soil conditions. Therefore, more stands need to be studied.

In the Guangdong Province of China, afforestation efforts have increased the forest cover from approximately 20% in the 1950s to approximately 60% at present (*Zhou et al., 2008*). *Schima superba* is a native tree species in south China, and plantations cover up to 1,236 hm$^2$ in Guangdong Province (*Hu et al., 2007*). In 2008, a study on *S. superba* water relations was conducted in the South China Botanical Garden. Previous studies compared seasonal hydraulic characters of *S. superba,* including sap flux density, whole-tree transpiration, and hydraulic conductance (*Mei et al., 2010a*; *Zhu et al., 2010*; *Zhang et al., 2014*; *Zhou et al., 2015*). However, at the individual-scale, we lack an understanding of drought stress strategy, and coordinated regulation of transpiration via stoma and hydraulic conductance. At the stand-scale, we have not yet quantified the effects of rainfall on annual and seasonal patterns of *S. superba* water use, or even the annual consumption of water.

As one of the major determinants of the rate of water flux through trees at the stand-scale, many studies have evaluated $g_c$, mainly using the JS modeling approach (*Martin et al., 1997*; *Magnani et al., 1998*; *Zhang, He & Assmann, 2008*) and PM equations ((*Morris, Mann & Collopy, 1998*; *Granier, Biron & Lemoine, 2000a*; *Harris et al., 2004*); Komatsu et al. 2006; *Whitley et al., 2009*). However, less work was done in comparing inter-class canopy conductance.

In this study, our objectives were to: (1) quantify water use for each tree rank and the plantation as a whole; (2) determine iso/anisohydry; (3) quantify inter-class canopy conductance and verify the possibility of a mean instantaneous conductance substitute for canopy conductance in the same canopy layer; and (4) determine if stomatal conductance determines water use more than hydraulic conductance.

**Table 1  Tree form features of 21 sampled trees. All trees were measured in April, 2011.**

| Tree No. | Diameter at breast height ($DBH$/m) | Height (m) | Canopy diameter (m*m) | Sapwood area ($A_s$/m$^2$) | Leaf area ($A_L$/m$^2$) |
|---|---|---|---|---|---|
| 1 | 0.15 | 15.30 | 6.4 × 2.3 | 0.016 | 66.97 |
| 2 | 0.19 | 12.60 | 6.7 × 4.3 | 0.025 | 101.83 |
| 3 | 0.13 | 12.10 | 4.5 × 2.3 | 0.012 | 54.16 |
| 4 | 0.22 | 15.30 | 6.6 × 5.6 | 0.031 | 125.66 |
| 5 | 0.22 | 15.50 | 6.9 × 5.3 | 0.032 | 129.50 |
| 6 | 0.10 | 11.00 | 1.2 × 0.9 | 0.007 | 30.86 |
| 7 | 0.18 | 12.90 | 5.5 × 5.0 | 0.020 | 85.71 |
| 8 | 0.09 | 9.70 | 3.4 × 3.9 | 0.006 | 27.15 |
| 9 | 0.09 | 9.50 | 2.3 × 2.6 | 0.006 | 27.15 |
| 10 | 0.24 | 16.90 | 7.0 × 6.2 | 0.036 | 145.34 |
| 11 | 0.14 | 11.20 | 2.9 × 4.3 | 0.013 | 55.53 |
| 12 | 0.07 | 8.00 | 2.0 × 2.6 | 0.004 | 16.36 |
| 13 | 0.08 | 12.00 | 3.1 × 1.8 | 0.006 | 25.12 |
| 14 | 0.14 | 13.10 | 4.4 × 3.1 | 0.014 | 61.86 |
| 15 | 0.07 | 9.70 | 2.4 × 1.8 | 0.004 | 19.86 |
| 16 | 0.19 | 13.70 | 4.6 × 4.8 | 0.025 | 101.83 |
| 17 | 0.21 | 15.40 | 7.8 × 4.6 | 0.029 | 118.11 |
| 18 | 0.20 | 14.90 | 5.4 × 5.8 | 0.026 | 108.04 |
| 19 | 0.14 | 11.20 | 4.1 × 4.9 | 0.014 | 59.72 |
| 20 | 0.15 | 13.40 | 4.4 × 4.5 | 0.015 | 64.76 |
| 21 | 0.26 | 13.90 | 6.4 × 6.8 | 0.041 | 164.03 |

# MATERIALS & METHODS

## Site description

The experiments were conducted in a *S. superba* plantation (23°10′N, 113°21′E, 41.4 m alt) that was planted in the mid-1980s and located in the ecological observation station of the South China Botanical Garden, Chinese Academy of Sciences, Guangzhou. The study area had a subtropical monsoon climate, with an annual mean temperature of 21.8 °C, and mean temperature of 32.7 °C in the hottest month of July, and 9.8 °C in the coldest month of January. Annual mean rainfall was 1750 mm, with April to September tending to be wet, and October to March of the next year tending to be dry. In 2010, the annual mean rainfall was 2148.4 mm and in 2011, annual mean rainfall was 1421.2 mm (*Guangzhou Bureau of Statistics, 2010*; *Guangzhou Bureau of Statistics, 2011*). The soil was loamed with an organic content of 2.3% and total nitrogen content of 0.07%. This *S. superba* plantation was planted at a density of 603 plants/hm$^2$, and the mean height of the stand at the time of the study was 12.7 m. The experimental site covered approximately 2885.6 m$^2$.

## Tree architecture characteristics

Characteristics of the sampled trees were measured in April 2011 to avoid age effects on transpiration (Table 1) (*Zhang et al., 2014*). Trees 1–15 were used to estimate transpiration and were divided into four groups based on diameter at breast height (DBH) (Table 2)
**Table 2  Features of plot and sample trees used in classification.** DBH standed for diameter at breast height. Number of trees (I) showed tree numbers of stand. Number of trees (II) showed sampling tree numbers for evaluated stand water use. Number of trees (III) showed sampling tree numbers for measuring leaf water potential and instantaneous stomatal conductance (Tree No. 17 ∼18, 21).

| | Total | Rank 1<br>DBH > 0.20 m | Rank 2<br>0.15 m < DBH ⩽ 0.20 m | Rank 3<br>0.10 m < DBH ⩽ 0.15 m | Rank 4<br>0.05 m < DBH ⩽ 0.10 m |
|---|---|---|---|---|---|
| Plot (2,885.6m$^2$) | | | | | |
| Number of trees (I) | 174 | 14 | 48 | 71 | 41 |
| Sample trees | | | | | |
| Number of trees (II) | 15 | No. 4, 5, 10 | No.1, 2, 7 | No.3, 11, 14 | No. 6, 8, 9, 12, 13, 15 |
| $A_s$ (m$^2$) | 0.081 | 0.037 | 0.023 | 0.015 | 0.006 |
| Projected canopy area | 15.85 | 7.65 | 4.65 | 2.39 | 1.16 |
| Number of trees (III) | 6 | No.17, 21 | No.16, 18 | No.19, 20 | |
| $A_s$ (m$^2$) | 0.028 | 0.040 | 0.028 | 0.016 | |

Trees 16–21 were used to measure leaf water potential and hydraulic conductance, and were classified into three groups according to DBH (Table 2). Trees 17–18 and 21 were used to analyze stomatal conductance. Tree height (H, m) was measured using a clinometer (Suunto Oy, Vantaa, Finland). DBH was measured using a diameter tape. Cores from the stems of five trees outside and close to the plot were taken using an increment core borer. The sapwood depth was measured using a caliper as sapwood and heartwood in *S. superba* are easy to identify visually. An exponential regression (see Eqs. (3-1), (2) and (3) below) between the sapwood area ($A_s$, m$^2$) and DBH was established based on the above measures, which was used to calculate the $A_s$ of the 15 trees in the plot. Similarly, leaf area ($A_L$) was calculated using a log function (see (6) below) established from the five trees outside and close to the plot.

### Environmental measurements

We continuously monitored wind speed (WS, m s$^{-1}$), air temperature (TA, °C), relative humidity (RH, %), and photosynthetically active radiation (PAR, μmol m$^{-2}$ s$^{-1}$) above the *S. superba* forest canopies from an observation tower set up within the plantation. Soil moisture (SM, m$^{-3}$ m$^{-3}$) was measured 30 cm below the ground surface from October 2010 until December 2011. Wind speed was measured using cup anemometers (AN4; Delta-T Devices, UK). TA and RH were measured using a temperature and humidity sensor (RHT2V-418; Delta-T Devices, UK). PAR was measured using a Li-Cor quantum sensor (LI-190SA; LI-COR, USA). SM was measured using three frequency domain sensors (SM200; Delta-T Devices) at a depth of 30–40 cm, which were set in a triangle around trees 17–21. The output values were measured every 30 s, and 10-min mean values were logged using a DL2e Delta-T logger (DL2e; Delta-T Devices).

The vapor pressure deficit (VPD, kPa) combined with the parameters TA and RH was calculated using the following formula (*Campbell & Norman, 1998*):

$$VPD = ae^{\left(\frac{bTA}{TA+c}\right)}(1-RH) \tag{1}$$

Where, a, b, and c are fixed parameters (0.611 kPa, 17.502 [unitless], and 240.97 °C, respectively).

## Sap flux density measurements

Sap flux density ($J_s$) was continuously measured from 2010 until 2011 using a homemade thermal dissipation probe (TDP). The Grainer-type probes were inserted radially at a depth of 20 mm into the stem of 21 trees at 1.3 m aboveground, one pair for each tree. The upper heated probe was supplied with a constant DC current (120 mA). The unheated probe was installed 10–15 cm below the upper sensors. The temperature difference between the paired sensors was recorded using the same logger as environmental measurements. All probes were installed on the north-facing side of the trees and covered with a plastic case, and shielded using a radiation-insulating film in order to minimize the possible effect of direct sunshine.

Sap flux density ($J_s$, g m$^{-2}$ s$^{-1}$) was weighted using the following equation:

$$J_s = \lambda(40 * J_{0-40} + (d-40) * J_{40})/d(d > 40) \, or J_{0-40}(d \leqslant 40) \tag{2}$$

where, $J_{0-40}$ (Sap flux density at 0–40 mm) is calculated using the empirical equation proposed by *Granier (1987)*, d is sapwood thickness, and $\lambda$ are correction factors of 1.164, 1.128, and 1.096 for Ranks 1, and 3, respectively. $J_{40}$ (sap flux density at $\geqslant$40 mm) was calculated following *Mei et al. (2010b)*:

$$J_{40} = 0.45 \times J_{0-40} \tag{2-1}$$

Whole-tree transpiration ($F_s$, g d$^{-1}$) was calculated as:

$$F_s = \sum(J_{0-40} \times A_{0-40} + J_{40} \times A_{40}) \times t \tag{3}$$

where, $t$ is 600 s (data were averaged and stored every 10 min in the logger), $A_{0-40}$ and $A_{40}$ are the sapwood areas in the outer xylem (0–40 mm) and inner xylem ($\geqslant$40 mm), respectively, which were calculated as:

$$A_{0-40} = 0.166 \bullet (DBH)^{1.336} \tag{3-1}$$
$$A_{40} = A_s - A_{0-40} \tag{3-2}$$

where $A_s$ is the total sapwood area ($m^2$), calculated using the regression equation relating it to $DBH$ (m) as follows:

$$A_s = 0.465 \bullet (DBH)^{1.794}. \tag{3-3}$$

Monthly water use ($\sum F_s$, g mon$^{-1}$) was calculated as:

$$\sum F_s = 30 \times F_s \tag{4}$$

Monthly canopy transpiration ($\sum E_L$, g m$^{-2}$ mon$^{-1}$) was calculated as:

$$\sum E_L = 30 \times E_L = 30 \times \frac{F_s}{A_L} \tag{5}$$

Where, $E_L$ is canopy transpiration (g m$^{-2}$ d$^{-1}$), $A_L$ is the total leaf area (m$^2$), which was calculated using a regression equation relating it to DBH (m) as follows (*Schäfer, Oren & Tenhunen, 2000*):

$$\log A_L = 1.672 \bullet \log DBH + 3.199 \tag{6}$$

Annual stand transpiration ($E_s$, mm y$^{-1}$) was calculated from $\sum\sum F_s$ per ground area over 1 year. $\Delta E_L$ was the difference between predawn and midday $E_L$

## Leaf water potential and stomatal conductance

The branches sampling procedure was conducted on trees 16–21 over 5–7 continuous sunny days in five time periods, Oct 2010, Jan, Apr, Jul, and Oct 2011. Leaf water potential ($\psi_L$, MPa) was measured using a portable pressure chamber in two ways: at 1-hour intervals from 05:00–06:00 to 20:00 in Oct 2010, and point measuring at 05:00–06:00, 13:00, and 20:00 in the other time periods. This minimized the destructive sampling of tree branches. Predawn leaf water potential ($\psi_{pd}$, MPa) and midday leaf water potential ($\psi_{md}$, MPa) referred to $\psi_L$ at 05:00–06:00 and 13:00, respectively. Iso/anisohydraulic regimes during the time periods were determined using the linear framework of (*Martinez-Vilalta et al., 2014*):

$$\psi_{md} = \Lambda + \sigma \bullet \psi_{pd} \tag{7}$$

Where $\Lambda$ is the intercept of the relationship, and $\sigma$ is the slope. A $\sigma = 0$ implies strict isohydry, $\sigma = 1$ implies strict anisohydry, $\sigma 1$ implies extreme anisohydry and $0 < \sigma < 1$ implies partial isohydry.

Instantaneous stomatal conductance ($g_s$, m s$^{-1}$) was simultaneously measured on trees 17–18 and 21 using a Li-6400 Portable Photosynthesis System, at the same time intervals and periods as $\psi_L$. $\Delta g_s$ represented the difference between predawn and midday $g_s$. All measurements were taken three times.

## Canopy conductance and whole-tree hydraulic conductance

Canopy conductance ($g_c$, m s$^{-1}$) was calculated by inverting the following equation (*Lu et al., 2003*):

$$\lambda E_c = \frac{\delta R_n + K_t \rho C_p \text{VPD} g_a}{\lambda [\delta + \gamma (1 + g_a/g_c)]} \tag{8}$$

where, $\lambda$ is the latent heat of vaporization (2.39 MJ kg$^{-1}$), $E_c$ (mm h$^{-1}$) is estimated from whole-tree transpiration divided by the projected canopy area of 3.96 m$^2$, $\delta$ is the slope of the relationship between saturated vapor pressure and temperature (kPa °C$^{-1}$), $R_n$ is the net radiation above the forest canopy (MJ m$^{-2}$ h$^{-1}$), $K_t$ is a unit conversion equal to 3,600 s h$^{-1}$, $\rho$ is air density (1.29 kg m$^{-3}$), $C_p$ is the specific heat of air (1.013 MJ kg$^{-1}$ °C$^{-1}$), $\gamma$ is the psychometric constant (0.066 kPa °C$^{-1}$), and $g_a$ is aerodynamic conductance (m s$^{-1}$), which was calculated using the method in *Delzon et al. (2004)*.

Whole-tree sapwood-specific hydraulic conductance $K_p$ and leaf-specific hydraulic conductance $K_L$ (g m$^{-2}$ s$^{-1}$ Mpa$^{-1}$) were indirectly estimated following *Cochard, Bréda & Granier (1996)*:

$$K_p = J_d/(\psi_{pd} - \psi_{md}) \quad \text{or} \quad K_L = J_l/(\psi_{pd} - \psi_{md}) \tag{9}$$

where, $J_d$ and $J_l$ are the difference in sap flux density of sapwood and leaf area between predawn and midday (g m$^{-2}$ s$^{-1}$), respectively. We ignored time lags between sap flux density and transpiration (*Zhao et al., 2016*).

## Data analysis

January and July represented the dry and wet seasons, respectively, and October 2010 and 2011 represented annual differences in south China, respectively. All statistical analyses

were performed using SPSS 15.0 (SPSS, Chicago, IL, USA) and Origin 8.1 (OriginLab, Northampton, MA, USA). The paired-samples $t$-test was used to analyze the seasonal and annual differences in $K_p$ and $K_L$. Major factors affecting $E_L$ and $F_s$ were determined using multilinear regression. All data were standardized before modeling. The strengths of relations were evaluated using the absolute values of the coefficients of separate models.

## RESULTS

### Effects of tree size on daily sap flux density, whole-tree and canopy transpiration

$J_s, F_s,$ and $E_L$ varied month by month, although all were at their maximums in the wet season and at their minimums in the dry season. The inter-class daily maximum of $J_s$ followed weakly their size in 2010 and 2011 (Figs. 1A–1B). The total occurring rate of $J_s$ ranking of Ranks 1 and 4 mapping their size ranking were 33.3% and 25.0%, respectively, which were higher than 4.2% and 12.5% of trees ranked 2 and 3 during the 2 years. Moreover, the separate mapped rate for Ranks 1-4 in 2010 were higher than that in 2011 except Rank 2, i.e., 41.7%, 0%, 16.7% and 33.3% compared with 25.0%, 8.3%, 8.3% and 16.7%, in turn. The total and separate mapped rates on $E_L$ for the four Ranks were similar to that of $J_s$ (Figs. 1C–1D). In contrast, the mapped rates on $F_s$ increased up to 100% and 100% in Ranks 1 and 4, respectively, and up to 58.3% and 58.3% in Ranks 2 and 3, respectively (Figs. 1E–1F). Meanwhile, the separate mapped rates for all the four Ranks in 2010 were no different from that in 2011. $F_s$ was evidently more close to present the assumption of the larger the tree, the greater the transpiration among the three parameters.

### Seasonal patterns of rank water use and whole-tree transpiration

The $\sum F_s$ and $\sum E_L$ of trees ranked 1–4 generally showed a low-high-low trend, which followed the change in rainfall over the dry season (one to three months)-wet season (four to nine months)-dry season (10–12 months) (Figs. 2A–2B). The mean annual $\sum F_s$ for Ranks 1, 2, 3, and 4 were 1301.28, 472.69, 469.28, and 135.34 g mon$^{-1}$ in the wet season, and 924.25, 348.05, 345.53, and 106.65 g mon$^{-1}$ in the dry season. The mean annual $\sum E_L$ for Ranks 1, 2, 3, and 4 were 9.75, 5.57, 8.21, and 5.54 g m$^{-2}$ mon$^{-1}$ in the wet season, and 6.92, 4.10, 6.04, and 4.37 g m$^{-2}$ mon$^{-1}$ in the dry season. $\sum F_s$ and $\sum E_L$ of all tree ranks in the wet season were averagely 1.37 and 1.35 times higher than in the dry season during 2010 and 2011. That is, water use in the fast growth period was almost always more than 50%.

### Annual patterns of stand water use

The total water use for Ranks 1, 2, 3, and 4 were 312.8, 185.9, and 126.9 mm in the wet season, and 262.4, 145.9 and 116.5 mm in the dry season in 2010. The mean daily $E_s$ was 0.86 mm d$^{-1}$ in 2010 and 0.72 mm d$^{-1}$ in 2011. Moreover, $F_s$ of all ranks declined to a certain extent, i.e., Ranks 1 and 2 declined from 40.2 and 15.8 g d$^{-1}$ in 2010 to33.0 and 11.2 g d$^{-1}$ in 2011, respectively, while Ranks 3 and 4 declined from 13.9 and 4.0 g d$^{-1}$ in 2010 to 12.9 and 3.9 g d$^{-1}$ in 2011, respectively.

The stand $\sum F_s$ was determined mainly by soil moisture ($SM$) and rainfall, with the effect of $SM$ greater than that of rainfall (Table 3). Of the five factors ($PAR$, $VPD$, $SM$,
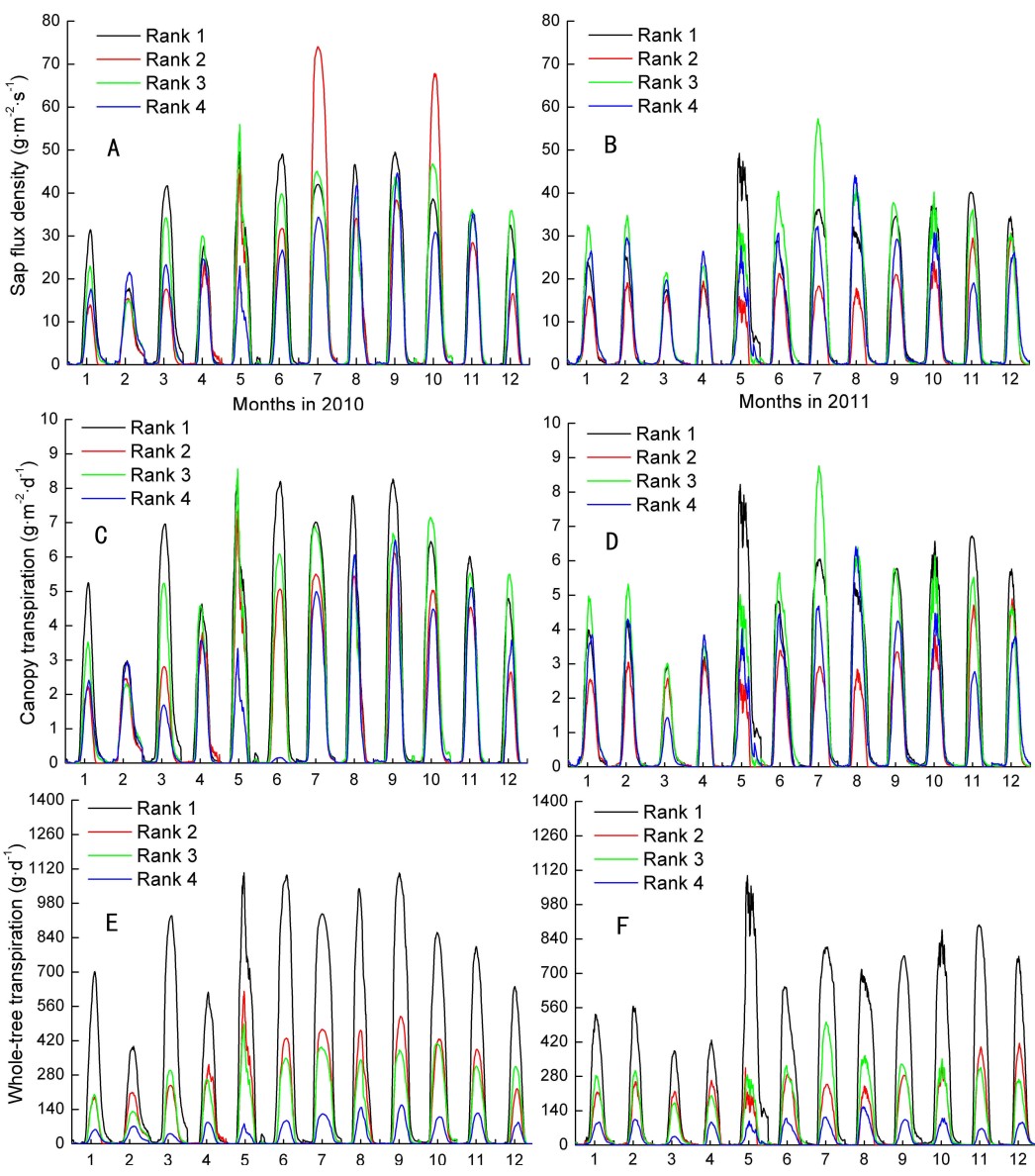

**Figure 1** **Inter-classes comparison in sap flux density, canopy transpiration and whole-tree transpiraiton.** Rank 1 represented trees at DBH > 0.20 m; Rank 2 was trees at 0.15 m< DBH ≤ 0.20 m; Rank 3 was trees at 0.10 m< DBH ≤ 0.15 m; Rank 4 represented trees at 0.05 m< DBH≤ 0.10 m.

*WS*, and rainfall), only *SM* was significant in the linear regression. Of the remaining four factors, rainfall was more significant than *WS* in the linear regression. Of *PAR*, *VPD*, and *WS*, no significant relations were observed. However, $\sum E_L$ was determined mainly by *SM*, *PAR*, and rainfall, with the effect of *SM* >*PAR* >Rainfall >*WS* (Table 3). No relations were observed between $\sum E_L$, *VPD*, and *WS*.
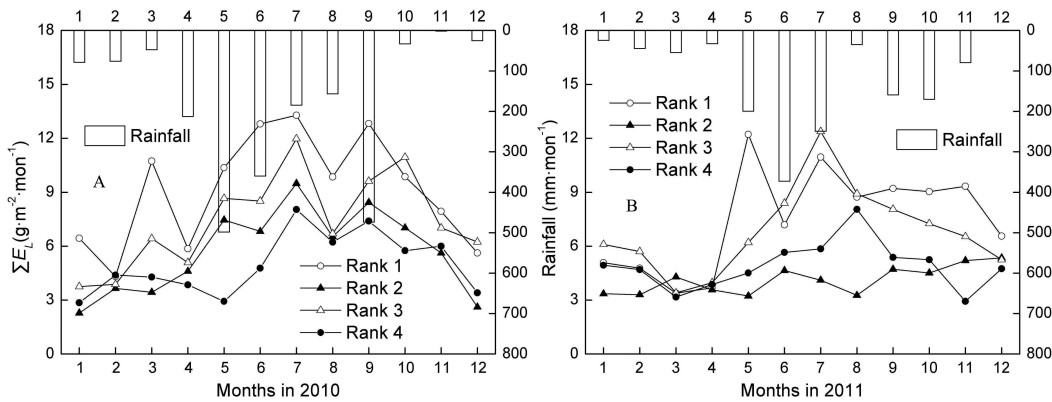

**Figure 2 Changes of monthly canopy transpiration and rainfall in 2010 and 2011.** Rank 1 represented trees at DBH > 0.20 m; Rank 2 was trees at 0.15 m < DBH ≤ 0.20 m; Rank 3 was trees at 0.10 m < DBH ≤ 0.15 m; Rank 4 represented trees at 0.05 m < DBH ≤ 0.10 m. $\sum E_L$ was monthly canopy transpiration.

**Table 3 Mutilinear regression analysis for $F_s$, $E_L$, $\Delta E_L$, PAR, VPD, SM, WS, Rainfall, $g_s$ and $K_L$.**

| Time | Depentent variable | n | Entered variables | Model | $R^2$ | P |
|------|--------------------|---|-------------------|-------|-------|---|
| 2011 | $F_s$ | 15 | PAR, VPD, SM, WS, Rainfall | $y = 3.65 \times 10^{-16} + 0.676 * SM$ | 0.46 | 0.016 |
| 2011 | $F_s$ | 15 | PAR, VPD, WS, Rainfall | $y = -2 \times 10^{-16} + 0.573 * Rainfall - 0.515 * WS$ | 0.60 | 0.016 |
| 2011 | $F_s$ | 15 | PAR, VPD, Rainfall | $y = -9 \times 10^{-17} + 0.579 * Rainfall$ | 0.34 | 0.049 |
| 2011 | $E_L$ | 15 | PAR, VPD, SM, WS, Rainfall | $y = -1 \times 10^{-17} + 0.739 * SM$ | 0.55 | 0.006 |
| 2011 | $E_L$ | 15 | PAR, VPD, WS, Rainfall | $y = -6 \times 10^{-16} + 0.512 * PAR + 0.498 * Rainfall$ | 0.59 | 0.019 |
| 2011 | $E_L$ | 15 | PAR, VPD, WS/PAR, VPD | $y = -6 \times 10^{-16} + 0.588 * PAR$ | 0.35 | 0.044 |
| 2011 | $E_L$ | 15 | VPD, WS, Rainfall | $y = -6 \times 10^{-16} + 0.571 * Rainfall - 0.54 * WS$ | 0.62 | 0.012 |
| Oct 2010 | $\Delta E_L$ | 15 | $K_L$, $\Delta g_s$ | $y = -2.04 \times 10^{-16} + 0.874 * g_s - 0.282 * K_L$ | 0.77 | 0.000 |
| Jan 2011 | $\Delta E_L$ | 15 | $K_L$, $\Delta g_s$ | $y = -1 \times 10^{-16} - 0.021 * g_s - 0.678 * K_L$ | 0.47 | 0.023 |
| Jul 2011 | $\Delta E_L$ | 13 | $K_L$, $\Delta g_s$ | $y = 3.23 \times 10^{-16} - 0.284 * g_s + 0.721 * K_L$ | 0.48 | 0.038 |
| Oct 2011 | $\Delta E_L$ | 15 | $K_L$, $\Delta g_s$ | $y = -3 \times 10^{-17} - 0.079 * g_s - 0.633 * K_L$ | 0.42 | 0.037 |

**Notes.**
$F_s$ standed for whole-tree transpiration; $E_L$ was canopy transpiration; $\Delta E_L$ was the difference between predawn and midday $E_L$; PAR standed for photosynthetically active radiation; VPD represented vapor pressure deficit; SM was soil moisture; WS was wind speed; $g_s$ represented stomatal conductance; $\Delta g_s$ represented the difference between predawn and midday $g_s$; $K_L$ represented leaf-specific hydraulic conductance; $R^2$ was the coefficient of determination; $p$ was significance. $p < 0.1$, $p < 0.05$, and $p < 0.01$ standed for a significant, remarkable and very significant difference, respectively.

## Leaf water potential

Significant seasonal differences were observed in the $\psi_{md} - \psi_{pd}$ for Ranks 1, 2 and 3 (Table 4). Daily $\psi_{pd}$ and $\psi_{md}$ ranged from −0.45 to −0.20 MPa and −0.97 to −0.61 MPa in Jan with SM at 23.39∼23.91 m$^{-3}$ m$^{-3}$, and −0.25 to −0.16 MPa and −1.47 to −0.68 MPa in Jul with SM at 34.57∼35.03 m$^{-3}$ m$^{-3}$ in 2011 (Fig. 3A). However, no significant annual differences were found in the $\psi_{md} - \psi_{pd}$ for the three ranks (Table 4). Daily $\psi_{pd}$ ranged from −0.42 to −0.18 MPa and −0.40 to −0.20 MPa, and the $\psi_{md}$ ranged from −1.27 to −0.35 MPa and −1.33 to −0.56 MPa in the Oct 2010 and 2011, respectively.

The maximum of $\psi_L$ (actual water potential at predawn) was inversely proportional to the DBH ($n = 6$, $R^2 = 0.80$, $p < 0.05$), whereas no relation was found between the

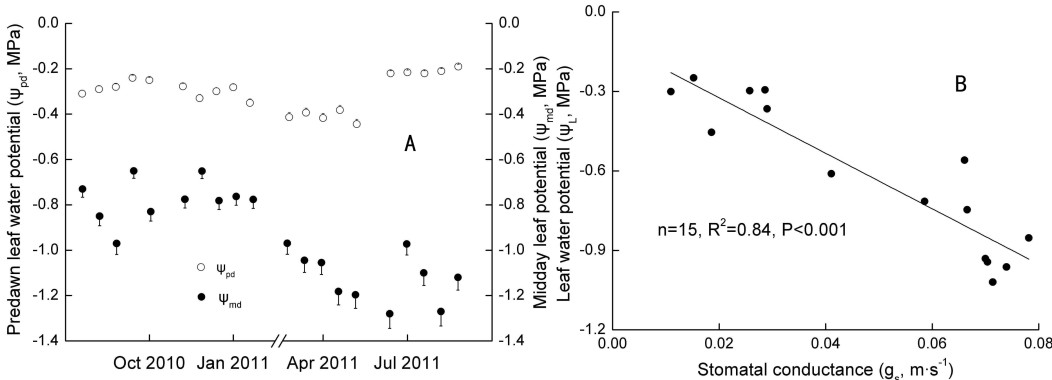

**Figure 3** **Seasonal changes and Linear regression of predawn and midday water potential.** $R^2$ was the coefficient of determination; $p$ was significance. $p < 0.1$, $p < 0.05$, and $p < 0.01$ standed for a significant, remarkable and very significant difference, respectively.

**Table 4** **Paired samples $t$ Test for $\psi_{md} - \psi_{pd}$, $K_p$ and $K_L$.** $\psi_{md}$ was midday leaf water potential; $\psi_{pd}$ was predawn water potential; $K_p$ was sapwood-specific hydraulic conductance; $K_L$ was leaf-specific hydraulic conductance. $SD$ was standard deviation; $df$ was degrees of freedom; $p$ was significance. $p < 0.1$, $p < 0.05$, and $p < 0.01$ represented a significant, remarkable and very significant difference, respectively.

| Test | $n$ | Annual difference | | | | | | | Seasonal difference | | | | | | |
|---|---|---|---|---|---|---|---|---|---|---|---|---|---|---|---|
| | | Oct 2010 | | Oct 2011 | | $t$ | $df$ | $p$ | Jan 2010 | | Jul 2011 | | $t$ | $df$ | $p$ |
| | | Mean | SD | Mean | SD | | | | Mean | SD | Mean | SD | | | |
| $\psi_{md} - \psi_{pd}$ in Rank 1 | 10 | −0.58 | 0.30 | −0.62 | 0.27 | | | | −0.42 | 0.11 | −1.09 | 0.17 | | | |
| Result | | | | | | −0.52 | 9 | 0.614 | | | | | 9.68 | 9 | 0.000 |
| $\psi_{md} - \psi_{pd}$ in Rank 2 | 10 | −0.58 | 0.24 | −0.58 | 0.14 | | | | −0.50 | 0.13 | −0.98 | 0.22 | | | |
| Result | | | | | | −0.01 | 9 | 0.996 | | | | | 6.30 | 9 | 0.000 |
| $\psi_{md} - \psi_{pd}$ in Rank 3 | 10 | −0.52 | 0.27 | −0.52 | 0.14 | | | | −0.38 | 0.04 | −0.76 | 0.21 | | | |
| Result | | | | | | −0.10 | 9 | 0.956 | | | | | 5.82 | 9 | 0.000 |
| $K_p$ in Rank 1 | 10 | 59.91 | 17.40 | 57.84 | 13.93 | | | | 58.49 | 13.32 | 39.33 | 14.88 | | | |
| Result | | | | | | 0.27 | 9 | 0.793 | | | | | 1.90 | 9 | 0.021 |
| $K_p$ in Rank 2 | 10 | 70.69 | 23.40 | 44.37 | 10.84 | | | | 83.75 | 31.84 | 23.40 | 8.67 | | | |
| Result | | | | | | 2.66 | 9 | 0.026 | | | | | 4.65 | 9 | 0.000 |
| $K_p$ in Rank 3 | 10 | 86.57 | 53.04 | 37.00 | 15.17 | | | | 73.58 | 43.71 | 62.77 | 25.85 | | | |
| Result | | | | | | 3.70 | 9 | 0.005 | | | | | 0.23 | 9 | 0.614 |
| $K_L$ in Rank 1 | 10 | 1.67 | 0.48 | 1.62 | 0.40 | | | | 1.64 | 0.37 | 3.36 | 1.26 | | | |
| Result | | | | | | 0.25 | 9 | 0.811 | | | | | −4.28 | 9 | 0.002 |
| $K_L$ in Rank 2 | 10 | 1.92 | 0.64 | 1.20 | 0.29 | | | | 2.28 | 0.87 | 1.34 | 0.71 | | | |
| Result | | | | | | 2.66 | 9 | 0.026 | | | | | 4.24 | 9 | 0.002 |
| $K_L$ in Rank 3 | 10 | 2.23 | 1.37 | 0.95 | 0.39 | | | | 1.89 | 1.13 | 3.12 | 1.25 | | | |
| Result | | | | | | 3.69 | 9 | 0.005 | | | | | −1.77 | 9 | 0.110 |

**Table 5   Linear regression for $g_s$ and $K_L$.** $g_s$ was stomatal conductance; $K_L$ was leaf-specific hydraulic; $R^2$ was the coefficient of determination; $p$ was significance. $p < 0.1$, $p < 0.05$, and $p < 0.01$ represented a significant, remarkable and very significant difference, respectively.

| Time | Depentent variable | $n$ | Independent variable | $R^2$ | $p$ |
|------|-------------------|-----|---------------------|-------|-----|
| Oct 2010 | $g_s$ in Rank 1 | 15 | $g_c$ | 0.67 | <0.001 |
| Jan 2011 | $g_s$ in Rank 1 | 15 | $g_c$ | 0.41 | <0.01 |
| Apr 2011 | $g_s$ in Rank 1 | 15 | $g_c$ | 0.38 | <0.05 |
| Jul 2011 | $g_s$ in Rank 1 | 15 | $g_c$ | 0.63 | <0.001 |
| Oct 2011 | $g_s$ in Rank 1 | 15 | $g_c$ | 0.85 | <0.001 |

minimum $\psi_L$ (actual water potential at midday) and DBH. That is, bigger trees had more water available in the rhizosphere. However, the strongest potential capacity to access soil water may not have been related to tree size at the time of measuring. According to the model relating $\psi_{pd}$ and $\psi_{md}$ proposed by *Martinez-Vilalta et al. (2014)*, extreme strict anisohydry ($\sigma > 1$) was only found in Rank 3 in Jan and Rank 2 in Apr in 2011 ($n = 10$, $R^2 = 0.54$ and 0.48, $p < 0.05$).

## Stomatal conductance and canopy conductance

Daytime hourly mean $g_s$ (trees 17, 18 and 21) ranged from 0.011 to 0.151 m s$^{-1}$. A linear relation between $g_s$ and *PAR* was observed in all five time periods. In contrast, a linear relation between $g_s$ and *VPD*, *SM*, and *WS* was only observed in Oct, Apr and Jul, Jan and Oct in 2011, respectively.

   Canopy conductance varied seasonally (much higher in the wet season than in the dry season), which was in agreement with *Kumagai et al. (2004)* and *Barradas et al. (2005)*. The $g_c$ for the whole plantation ranged from 0.00058–0.034 mm s$^{-1}$ in the wet season and 0.000092–0.070 mm s$^{-1}$ in the dry season. Individually, the $g_c$ of trees ranked 1–4 ranged from 0–0.00069 mm s$^{-1}$. Stomatal conductance of Rank 1 trees was positively related to $g_c$ in each of the five time periods (Table 5).

## Hydraulic conductance and canopy conductance

Daily $K_p$ and $K_L$ did not vary with tree size. A proportional relationship between $K_p$ or $K_L$ and *DBH* was observed in Oct 2011 ($n = 6$, $R^2 = 0.55$ and 0.64, $p < 0.1$). Moreover, both the maximums and minimums of $K_p$ and $K_L$ occurred in Rank 2 and 3 trees.

   There were significant differences in $K_p$ and $K_L$ for Rank 2 and 3 trees between 2010 and 2011 (Table 4). The $K_p$ and $K_L$ for Rank 2 trees in Oct 2010 was 70.69 and 86.57 g m$^{-2}$ s$^{-1}$ MPa$^{-1}$, respectively. The $K_p$ and $K_L$ for Rank 3 trees in Oct 2010 was 1.92 and 2.23 g m$^{-2}$ s$^{-1}$ MPa$^{-1}$, respectively. The $K_p$ and $K_L$ for Rank 2 trees in Oct 2011 was 44.37 and 36.40 g m$^{-2}$ s$^{-1}$ MPa$^{-1}$, respectively. The $K_p$ and $K_L$ for Rank 3 trees in Oct 2011was 1.20 and 0.95 g m$^{-2}$ s$^{-1}$ MPa$^{-1}$, respectively

   There were significant seasonal differences for Rank 1 and 2 trees in the 2 years (Table 4). The mean daily $K_p$ and $K_L$ for Rank 1 trees in the wet season was 39.33 and 23.40 g m$^{-2}$ s$^{-1}$ MPa$^{-1}$, respectively. The mean daily $K_p$ and $K_L$ for Rank 2 trees in the wet season was 0.032 and 0.013 g m$^{-2}$ s$^{-1}$ MPa$^{-1}$, respectively. The mean daily $K_p$ and $K_L$ for Rank

1 trees in the dry season was 57.84 and 44.37 g m$^{-2}$ s$^{-1}$ MPa$^{-1}$, respectively. The mean daily $K_p$ and $K_L$ for Rank 2 trees in the dry season was 0.016 and 0.012 g m$^{-2}$ s$^{-1}$ MPa$^{-1}$, respectively.

In addition, $\Delta E_L$ was well related to $\Delta g_s$ and $K_L$ in Oct 2010 and 2011, and related to $K_L$ in Jan and Jul 2011, respectively (Table 3). Daily $\Delta E_L$, $\Delta g_s$ and $K_L$ ranged from 0.65–10.02 g m$^{-2}$ d$^{-1}$, 0.01–0.20 m s$^{-1}$, and 0.0082–0.051 g m$^{-2}$ s$^{-1}$ Mpa$^{-1}$, respectively.

## DISCUSSION

### Scaling up water use from the individual to the stand

$F_s$ showed stronger relation with the ranks than $J_s$ and $E_L$. There are fewer studies comparing this parameter, but more interspecies studies comparing daily mean $J_s$. *Horna et al. (2011)* compared $J_s$ of seven species with no size classes in July and December between 2007 and 2008, but found no significant differences. Under a common setting criterion (i.e., depth of the xylem, azimuth), $J_s$ showed high interspecific differences (*Ewers et al., 2002*; *O'Brien, Oberbauer & Clark, 2004*; *Daley & Phillips, 2006*; *Horna et al., 2011*). Even within species, $J_s$ may vary with tree social position (*Ambrose et al., 2010*), cultural type (*Kunert et al., 2012*), local condition (*Granier et al., 1990*; *Motzer et al., 2005*), age (*Oguntunde & Oguntuase, 2007*), and tree-size parameters (*Loustau et al., 1996*; *Ewers et al., 2002*; *Kume et al., 2010*). Therefore, future research should sample a large number of trees to improve the accuracy (*Oishi, Oren & Stoy, 2008*). Many studies have provided evidence of the mechanism of this relation. Some studies suggested that variability of $J_s$ is related to changes in a single environmental factor (e.g., *Granier et al., 1990*), while others support the influence of multiple environmental factors (*Bovard et al., 2005*; *Daley & Phillips, 2006*). Many studies that used $F_s$ instead of $J_s$ might have concluded different sap flow measurements, which also likely amplified individual tree differences (*Roberts, Vertessy & Grayson, 2001*; *Tausend, Meinzer & Goldstein, 2000*; *Motzer et al., 2005*; *Yunusa et al., 2010*; *Zeppel et al., 2010*). We found that $F_s$ showed "the larger the tree, the greater the transpiration" more effectively than $J_s$.$E_L$ takes into account the importance of leaf area. Some studies have found that $E_L$ is an important determinant of tree water use (*Myers et al., 1996*; *Radersma, Ong & Coe, 2006*). In our study, clear differences in $E_L$ were also found between ranks, i.e., the leaf-area of Rank 1 was 1.48 times that of Rank 3, but transpired 51.6% more water in Jul 2011.

The seasonal decline in water use with leaf area has been shown in many studies. This has been explained as preventing canopy desiccation (*Dye, 1996*; *Farrington et al., 1994*; *Hutley, O'Grady & Eamus, 2000*). In this study, tree water use only accounted for 9.9% and 13.9% of rainfall in the 2010 and 2011 wet seasons, respectively, and 48.4% and 31.2% of rainfall in the 2010 and 2011 dry seasons, respectively. However, we only showed a decline in tree water use in the dry season after ignoring leaf growth. It showed the effects of external environmental factors more effectively than current approaches. *Llorens et al. (2010)* also showed that transpiration decreased significantly during the drier summers of 1998 and 2000, compared with the wetter summer of 1997.

Many studies have reported that $SM$ determines transpiration (*Jiao et al., 2015*; *Gazal et al., 2006*; *Chang et al., 2014*; *Zhao & Liu, 2010*). Some studies have observed a decline in

transpiration with decreasing *SM*. *Gartner et al. (2009)* found that birch and Norway spruce trees reduced their transpiration in response to drought. In this study, tree water use and canopy transpiration were significantly affected by *SM* at all daily and monthly scales. The decreasing transpiration could be partly attributed to the decrease in *SM* in the 2011 dry season, when rainfall decreased by 727.2 mm, compared with the 2010 wet season, although there were no *SM* data for 2010. Other studies have shown that rainfall influences tree transpiration when soil water in the upper profile is insufficient (*O'Grady, Eamus & Hutley, 1999*). Even in the afternoon, tree transpiration may vary before and after a rain event (*Wang et al., 2017*). Nevertheless, this effect did not show a simple proportional relation. *Huang & Zhang (2016)* reported that 0–5-mm precipitation increased transpiration, while >5mm precipitation decreased transpiration in two xerophytic shrubs. Conversely, water use was weakly related to rainfall, indicating that the trees strongly depended on groundwater (*Morris, Mann & Collopy, 1998*). The inconsistent effect of rainfall and *SM* was possibly related to root depth. *Zhao & Liu (2010)* showed that soil water content at 10–20-cm depth depended significantly on rainfall. It is likely that plants depend on water uptake up to 50 cm soil depth (*VanSplunder et al., 1996*; *Lagergren & Lindroth, 2002*). Our *SM* measurements at 30–40-cm depth determined the decrease in *S. superba* transpiration in the dry year (2011), which was possibly attributable to the combined effects of *SM* and rainfall. This was also shown in the decrease of 16.3% in $E_s$ between the two years, which was lower than 33.8% of precipitation.

The total water use of *S. superba* in 2010 and 2011 was approximately 14.6% and 18.5% of the average annual rainfall, respectively. The low level of water consumption was possible attributed to its growth near over-mature stage. From studies of broad-leaved plantations in south China, the total annual mean daily $E_s$ of 0.79 mm d$^{-1}$ of *S. superba* (30–35a) was higher than the 0.59 mm d$^{-1}$ of *Acacia mangium* (19a) at the Heshan experimental station in Guangdong Province (22textdegree40′N, 112°54′E, 226 m alt), and lower than the 1.01 mm d$^{-1}$ of *Eucalyptus urophylla* ×*E. grandis* (4–5a) at the Huangmian Forest Farm in Liuzhou, Guangxi Province (24°45.8′N, 109° 53.6′E, 80 m alt), and 1.48 and 1.53 mm d$^{-1}$ of *E. urophylla* at Hetou (21°05′N, 109°54′E, 25 m alt) and Jijia (20°54′N, 109°52′E, 70 m alt), respectively, in Leizhou Peninsula, Gouangdong Province (*Morris et al., 2004*; *Ma et al., 2008*; *Zhu et al., 2015*). As they have similar latitude/longitude and climate background, water use of plantations poses no threat to the ground water balance, irrespective of the mix of exotic and native species, although there are differences (*Myers et al., 1996*).

## Stomatal and hydraulic regulation of transpiration from the individual to the stand

Regulations of $g_s$ included physical factors, such as ABA, PH value, and flagellin (*Small & Maxwell, 1939*; *Zhang, He & Assmann, 2008*), and environmental factors, such as *VPD*, light intensity, air humidity, and atmospheric $CO_2$ concentration (*Jarvis, 1976*; *Aasamaa & Sõber , 2011*; *Monteith, 1995*). *Johnson et al. (2001)* reported that photosynthetic photon flux density (PPFD) was most correlated with $g_s$ in *Acer saccharum*. *Monteith (1995)* examined three phases of diurnal $g_s$, corresponding to diurnal *VPD*, and found that low *VPD* increased the minimal transpiration limited by stomata (phase C), and larger values of

*VPD* caused a large decline in $g_s$ (phase B), while minimal medium values of *VPD* increased transpiration (phase A). Although $g_s$ correlated well with *VPD* only in Oct 2011, our results followed *Monteith (1995)*.

Extreme anisohydry was found in *S. superba*, and that seemed to be a dynamic mechanism and varied with ranks and periods. However, it is doubtful that *S. superba*, which grew in South China, showed extreme anisohydry. *Martinez-Vilalta et al. (2014)* found only five species showed extreme anisohydry ($\sigma > 1$) in 102 species. Moreover, they noted that the phenomenon may occur in phreatophytes and drought-deciduous species. In our study, $\psi_{md}-\psi_{pd}$ was approximately maintained seasonally constant but $\psi_{pd}$ was correlated with soil water availability, which be in consistent with an isohydrodynamic behavior proposed by *Franks, Drake & Froend (2007)*. They suggested that the behavior was linked to a combined hydraulic regulation with stomatal control and plant hydraulic conductance. We also found there were annual and seasonal coordinated regulations with stoma and hydraulic conductance on canopy transpiration (Table 3). However, it was difficult to explain the difference between Oct 2010 and Jul 2011 only linking *SM* and rainfall, because the latter had higher *SM* and rainfall. It is still suggested that a higher *SM* in the dry season (compared with Jan and Oct 2011) partially resulted in stomatal inverse control on leaf water potential (Fig. 3B), even though the inverse control was not in agreement with $\psi_L$ positively controlling $g_s$ reported by *Comstock & Mencuccini (1998)*, *Williams & Araujo (2002)* and *Ripullone et al. (2007)*.

Actually, instantaneous $g_s$ did not effectively represent the bulk surface conductance unless it was sampled from different canopy layers. Stomatal conductance from the same canopy layer could be scaled up to calculate canopy stomatous conductance in our study (Table 5). It was merely too low values of $g_c$ in our study compared to those reported for Qinhai spruce (0.3–51.3 mm s$^{-1}$; (*Chang et al., 2014*)), Scots pine stands (13–28 mm s$^{-1}$; *Granier et al., 1996*; *Dolman et al., 1998*; *Sturm et al., 1998*), Norway spruce stands (10–13 mm$^{-1}$; *Lu et al., 1995*; *Alsheimer et al., 1998*; *Cienciala et al., 1998*), and *Fagus sylvatica* (3.3–18.5 mm$^{-1}$; *Magnani et al., 1998*). However, our results concurred with the ranges reported for *Prunus armeniaca* (0.0012–0.0024 mm s$^{-1}$; *Barradas et al., 2005*), and a mixed stand composed mainly of *Eucalyptus crebra* and *Callitris glaucophylla* (maximum 0.0083 mm s$^{-1}$; *Whitley et al., 2009*), and a *Pinus canariensis* forest (0.0033 mm s$^{-1}$; *Kučera et al., 2017*). The reported ranges of values vary greatly, not only by species (*Köstner et al., 1992*), age (*Forrester, Collopy & Morris, 2010*), temporal scales (*Bernier et al., 2006*), soil, root and canopy components (*Morris et al., 2004*), but also by different models of calculation using the same method with different parameters (e.g., *Morris, Mann & Collopy, 1998*; *Zeppel & Eamus, 2008*). In contrast, the inter-class daily conductance was smaller than that of whole trees, and the highest average conductance was 0.00069 mm s$^{-1}$. Moreover, the maximum $g_c$ always occurred in Rank 3 trees, as there were too few trees in some of the ranks, especially Rank 1. Thus, the results could not confirm that the larger trees determined the whole conductance (*Martin et al., 1997*).

## CONCLUSIONS

Plant-water relations of plantations have been increasingly studied, especially in south China, which has a large and growing proportion of plantation forests. The water consumption of *S. superba* trees showed no threat to the ground water balance. The annual and seasonal differences in water use were significantly affected by *SM* and rainfall. Stoma responded to physical hydraulic factors, such as water potential, showed an isohydrodynamic behavior, and were positively linearly related with external environmental factors such as *PAR*.

### Funding

This study was financially supported by the National Natural Science Foundation of China (Grant Nos. 41630752, 31670410) and the National Key Research and Development Programme (2016YFC0500106-02). The funders had no role in study design, data collection and analysis, decision to publish, or preparation of the manuscript.

### Grant Disclosures

The following grant information was disclosed by the authors:
National Natural Science Foundation of China: 41630752, 31670410.
National Key Research and Development Programme: 2016YFC0500106-02.

### Competing Interests

The authors declare there are no competing interests.

### Author Contributions

- Xiao-wei Zhao performed the experiments, analyzed the data, contributed reagents/materials/analysis tools, prepared figures and/or tables, authored or reviewed drafts of the paper, approved the final draft.
- Lei Ouyang authored or reviewed drafts of the paper, approved the final draft.
- Ping Zhao conceived and designed the experiments, performed the experiments, contributed reagents/materials/analysis tools, authored or reviewed drafts of the paper, approved the final draft.
- Chun-fang Zhang approved the final draft.

### Data Availability

    The raw data are provided in a Data S1.

### Supplemental Information

Supplemental information for this article can be found online at http://dx.doi.org/10.7717/peerj.5164#supplemental-information.

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
