# Peer review of "Effects of size and microclimate on whole-tree water use and hydraulic regulation in Schima superba trees"

_PeerJ, doi:10.7717/peerj.5164_

## Round 0.1 · original submission · Major Revisions

The submitted manuscript has been evaluated by three reviewers. All three reviewers think that the manuscript is of interest, but cannot be published at the present version. It reuires a major revision. Please revise carefully the manuscript incorporating the comments from three reviewers.

·

Basic reporting

no comment

Experimental design

no comment

Validity of the findings

no comment

Additional comments

The Schima superba plantation has been widely established in South China, mainly for the purpose of forest restoration. Using the TDP method, the authors aim to investigate seasonal water use patterns and hydraulic characteristics of four diameter size groups of a Schima superba plantation during 2010-2011. A novelty of this study is to improve our understanding for the water consumptions of this important plantation. I believe this paper will be of general interests to the readers of the journal, but can be improved significantly after revision. It would also be better to be edited by a native speaker. Below are comments:
1. Abstract. Line22 and 23, the authors mention ‘rank 3’, but without any information about this term. Add interpretations in Line 10 when the four ranks are first mentioned .
2. Introduction. In the first paragraph, I do not understand why the authors introduce the background of exotic vs native species plantations, because it is not related to the objectives of this study.
3. Line 81, 21 samples are selected for the study. But in Table 1, the total number is 15. I have a little confusion about the Table 1. It is too simple for the reader to obtain more specific information about the materials.
4. Line 84, ‘…..which avoided age effects…’ ? Should there be ‘season’ ?
5. In this study, all the sample trees are classified into four ranks according to their DBH. However, in Figure 1-5 and Figure 8, those tree individuals are classified into four groups as dominant, codominant, middle and depressed. Therefore, it is hard for readers to capture some key results.
6. Discussion. This section should be improved substantially. For example, in the first paragraph, they explain what is sap flux density (this should be put in introduction or method section), list a number of studies that compare inter- and intra-specific difference in Js, but talk less about their own results. Generally, the authors should first focus on the most important findings, followed by a mechanical explanation.

Reviewer 2 ·

Basic reporting

The topic of this paper is of great interest. The measurements seem to be well planed and carried out effectively resulting in a rich and valuable data set. With these data on plant sap flow and microclimate data, potentially interesting patterns can be identified that can be used to answer important scientific questions. A potentially great paper can be resulted from this work. However, I feel the paper need to be substantially improved before publication in the following aspects:
1) More detailed information need to be given in Materials and Methods regarding the number of trees used for each of the measurements. Need to tell how many trees in each of the rank are used for sap flow, water potentials and Kl. In case only a subset of trees were used for a specific measurement, explanations need to be given why not all the trees were measured.

2) The results need to be much better organized. The Table and Figures especially needs more works to be publishable. For example, in the caption/legends more detailed information need to be added to make it more easily understandable to readers. See more details below.

Introduction
Since the study is only about one native species, the mention of studies on exotic trees is not very relevant and confusing. Suggest not saying anything about exotic vs. native species in the Introduction.

The reasons given in line57-65 were not enough to justify the present study. Clear scientific questions or hypotheses need to be stated. For example, if you think the study between different DBH categories is important. You may expect them to be different in sap flow, water potential, KL, etc. You can then develop your hypotheses about the comparisons between DBH ranks.

Materials and Methods

Sap Flow measurements:
It is not totally clear how many trees were used for sap flow measurements. In line 111, it is said “we classified 15 trees into 4 groups” but in line 113-114 it is said 21 trees (instead of 15) were measured? What are the DBH ranges of trees No. 16-21? Why not also classify tree No. 16-21 into the four DBH categories rather than separate these trees from Tree No. 1-15?

Leaf water potential and stomatal conductance measurements (Line 134-144):
According to the description, trees No. 16-21 were measured and these trees were classified into the four DBH classes. How many trees in each of the four classes? If the purpose were to compare between different DBH classes, there would not be enough replicates (trees) for each of the classes.

Results
Overall need to me more concise and better organized to show the overall patterns. It would be better to show some of the major comparisons between Ranks in a table. Some of the important linear regressions are better shown graphically.

The quality of table and figures especially need to be improved:

Tables and figures need to be self-explanatory, e.g. all abbreviations need to be defined.

More detailed information need to be added to Table 1 to allow the readers understand your results, e.g. need to tell the tree identities (No. 1-21) within each of the four ranks. The DBH of the 21 sampled trees were measured and need to be reported in Table 1. Suggest to arrange and number the sample trees according to their DBH and tell in the table what kind of measurements were done for each of the tree.

All the figure legends are too short (only one short sentence) and do not provide sufficient information needed to understand your figures.

Figure 1&2: What do you mean by dominant, codominant, middle and depressed trees? The terms need to be defined in Materials and Methods. Do they correspond to tree Rank 1, 2, 3 and 4, respectively?

Fig. 4: According to M&M, 6 trees were used for water potential measurements. Are there 2 trees for each of the three classes? Two trees would not allow calculating standard errors. What are the error bars for in Fig. 4a?

Fig. 5. With these many data points, it is difficult to see the general patterns. Need some sort of simplification/summary to clearly show the pattern.

Fig. 6,7,8: Why only the specific Rank is shown but not other ranks?


Discussion

If I understand correctly, one of the major aims of the study is to compare plant water use of different DBH ranks. Thus, the discussions need to be more centered on the comparisons between the tree ranks.

Line 334-340: the discussion about Kl need to be expanded a bit. There is actually no real “discussion” about Kl but rather repeating some of the results.

Experimental design

no comment

Validity of the findings

no comment

Additional comments

The topic of this paper is of great interest. The measurements seem to be well planed and carried out effectively resulting in a rich and valuable data set. With these data on plant sap flow and microclimate data, potentially interesting patterns can be identified that can be used to answer important scientific questions. A potentially great paper can be resulted from this work. However, I feel the paper need to be substantially improved before publication in the following aspects:
1) More detailed information need to be given in Materials and Methods regarding the number of trees used for each of the measurements. Need to tell how many trees in each of the rank are used for sap flow, water potentials and Kl. In case only a subset of trees were used for a specific measurement, explanations need to be given why not all the trees were measured.

2) The results need to be much better organized. The Table and Figures especially needs more works to be publishable. For example, in the caption/legends more detailed information need to be added to make it more easily understandable to readers. See more details below.

Introduction
Since the study is only about one native species, the mention of studies on exotic trees is not very relevant and confusing. Suggest not saying anything about exotic vs. native species in the Introduction.

The reasons given in line57-65 were not enough to justify the present study. Clear scientific questions or hypotheses need to be stated. For example, if you think the study between different DBH categories is important. You may expect them to be different in sap flow, water potential, KL, etc. You can then develop your hypotheses about the comparisons between DBH ranks.

Materials and Methods

Sap Flow measurements:
It is not totally clear how many trees were used for sap flow measurements. In line 111, it is said “we classified 15 trees into 4 groups” but in line 113-114 it is said 21 trees (instead of 15) were measured? What are the DBH ranges of trees No. 16-21? Why not also classify tree No. 16-21 into the four DBH categories rather than separate these trees from Tree No. 1-15?

Leaf water potential and stomatal conductance measurements (Line 134-144):
According to the description, trees No. 16-21 were measured and these trees were classified into the four DBH classes. How many trees in each of the four classes? If the purpose were to compare between different DBH classes, there would not be enough replicates (trees) for each of the classes.

Results
Overall need to me more concise and better organized to show the overall patterns. It would be better to show some of the major comparisons between Ranks in a table. Some of the important linear regressions are better shown graphically.

The quality of table and figures especially need to be improved:

Tables and figures need to be self-explanatory, e.g. all abbreviations need to be defined.

More detailed information need to be added to Table 1 to allow the readers understand your results, e.g. need to tell the tree identities (No. 1-21) within each of the four ranks. The DBH of the 21 sampled trees were measured and need to be reported in Table 1. Suggest to arrange and number the sample trees according to their DBH and tell in the table what kind of measurements were done for each of the tree.

All the figure legends are too short (only one short sentence) and do not provide sufficient information needed to understand your figures.

Figure 1&2: What do you mean by dominant, codominant, middle and depressed trees? The terms need to be defined in Materials and Methods. Do they correspond to tree Rank 1, 2, 3 and 4, respectively?

Fig. 4: According to M&M, 6 trees were used for water potential measurements. Are there 2 trees for each of the three classes? Two trees would not allow calculating standard errors. What are the error bars for in Fig. 4a?

Fig. 5. With these many data points, it is difficult to see the general patterns. Need some sort of simplification/summary to clearly show the pattern.

Fig. 6,7,8: Why only the specific Rank is shown but not other ranks?


Discussion

If I understand correctly, one of the major aims of the study is to compare plant water use of different DBH ranks. Thus, the discussions need to be more centered on the comparisons between the tree ranks.

Line 334-340: the discussion about Kl need to be expanded a bit. There is actually no real “discussion” about Kl but rather repeating some of the results.

Reviewer 3 ·

Basic reporting

see detailed comments in pdf file.

Experimental design

see detailed comments in word file.

Validity of the findings

see detailed comments in word file.

Additional comments

see detailed comments in word file.

Annotated reviews are not available for download in order to protect the identity of reviewers who chose to remain anonymous.

---

## Round 0.2 · Minor Revisions

The revised manuscript has been evaluated by three reviewers. All reviews indicated that the authors had done a good job in revising. However, two of reviewers and me think it still needs a language check before acceptance, e.g., by an English service agency, to improve the writing. Please revise the manuscript as suggested. When resubmit, please submit one version with change-tracked, and one without change tracked

·

Basic reporting

no comment

Experimental design

no comment

Validity of the findings

no comment

Additional comments

The manuscript has been improved by the authors and deserves to be published in PeerJ. However, there are still several points that should be considered before publication.
1) This manuscript have many language problems, it would be better if edited by a native speaker who is an expert in water relations.
2) The ABSTRACT should be rewritten, I cannot find any key results of this study here, and the final conclusion makes no sense.
3) The DISCUSSION section, although improved, still needs extensive revisions. The authors should focus on their key results. This paper have six figures and six tables (too many).
4) The legends of the figures are too simple. Generally, we can understand the figure without reading the main text.
In figures 1-2, what is the rank 1-4? And it is not easy to distinguish different ranks by using the four symbols.
In figure 4B, change “maximum” to “minimum” if the water potential indicates the lowest value during a day.
In figure 5, no any explanations for figure 5A, and check the unit of stomatal conductance.
In figure 6, the symbols of the four ranks are different from figure 1-2.

Reviewer 2 ·

Basic reporting

The authors did a good job in revising their manuscript according to the comments from the three reviewers. I have no further concerns for the publication of this paper.

Experimental design

The authors did a good job in revising their manuscript according to the comments from the three reviewers. I have no further concerns for the publication of this paper.

Validity of the findings

The authors did a good job in revising their manuscript according to the comments from the three reviewers. I have no further concerns for the publication of this paper.

Additional comments

The authors did a good job in revising their manuscript according to the comments from the three reviewers. I have no further concerns for the publication of this paper.

Reviewer 3 ·

Basic reporting

see comments in word version

Experimental design

see comments in word version

Validity of the findings

see comments in word version

Additional comments

The author did lot of work on revising the manuscript which smooth the whole text substantially. However, I still have some concerns before acceptance.

Annotated reviews are not available for download in order to protect the identity of reviewers who chose to remain anonymous.

---

## Round 0.3 · accepted · Accept

The authors have revised the manuscript according to the comments from reviewers and the writing has been edited by language service agency so I have decided to accept it.

#